# forester: A Novel Approach to Accessible and Interpretable AutoML for Tree-Based Modeling

Anna Kozak[1]  Hubert Ruczyński[1]

[1]Warsaw University of Technology

**Abstract** The majority of AutoML solutions are developed in Python. However, a large percentage of data scientists are associated with the R language. Unfortunately, there are limited R solutions available with high entry level which means they are not accessible to everyone. To fill this gap, we present the *forester* package, which offers ease of use regardless of the user's proficiency in the area of machine learning.

The *forester* package is an open-source AutoML package implemented in R designed for training high-quality tree-based models on tabular data. It supports regression and binary classification tasks. A single line of code allows the use of unprocessed datasets, informs about potential issues concerning them, and handles feature engineering automatically. Moreover, hyperparameter tuning is performed by Bayesian optimization, which provides high-quality outcomes. The results are later served as a ranked list of models. Finally, the *forester* package offers a vast training report, including the ranked list, a comparison of trained models, and explanations for the best one.

## 1 Introduction

Machine learning is being used more and more in the world around us. Every day, models are created to assist doctors (Shimizu and Nakayama, 2020), financiers (Jorge et al., 2022), or tourists (Fararni et al., 2021). With the increasing demand for model building, research is being conducted on automatically developing tools to build artificial intelligence based solutions.

Many types of models are used in machine learning, such as decision rules (scoring card model) to complex neural network structures modeling natural language (large language models, for example, ChatGPT (Bavarian et al., 2022)). Viewing machine learning in terms of tabular data, we have a wide range of models available, from decision trees and linear or logistic regression to random forests, SVM, or neural networks. However, tree-based models are the most widely used; the main reason behind this is their high predictive efficiency. A simple decision tree model gives relatively satisfactory results, but using multiple trees to create a random forest allows significantly higher predictive power (Caruana et al., 2008; Grinsztajn et al., 2022).

Automating the process to build machine learning models can include many different components. For example, the CRoss Industry Standard Process for Data Mining (CRISP-DM) (Wirth and Hipp, 2000) is the most common methodology for data mining, analytics, and data science projects. But the basic framework of an automatic machine learning system is the preparation of models based on data entered by the user. This process can be extended in various directions; for example, a preliminary analysis of the given data can be taken care of to look for potential data errors or outlier observations, i.e. exploratory data analysis. Another essential element may be the search space of the model's hyperparameters. Optimization of hyperparameters can be based on simple methods such as a predefined parameter grid or random search. Another way to select hyperparameters is to use Bayesian optimization (Snoek et al., 2012) or meta-learning (Vilalta et al., 2004; Vanschoren, 2019; Woźnica and Biecek, 2022). After tuning the models with hyperparameter optimization, the next step we can add is to analyze the results in the form of a leaderboard

or visualization. By extending with explanatory methods (Biecek and Burzykowski, 2021) and reporting, the entire machine learning process can be finalized.

Automating the process of machine learning allows access to data science tools for people who are starting in data analysis and modeling. At the same time, it is an improvement and speeds up the work of experienced data scientists, who can make at least baseline models using a single line of code.

In this paper, we present the AutoML package written for the R (R Core Team, 2022) to create models for regression and binary classification tasks on tabular data. The main goals of the package are: making the package easy to use, fully automating all the necessary steps inside the ML pipeline, and providing results that are easy to create, understand and allow diagnostics of the models. The availability of responsible machine learning methods in the solution allows the results of complex models to be interpreted. Changing the focus from obtaining the best possible outcomes to the interpretability of the results is a novelty for the AutoML tools. The implementation of the *forester* package can be found in our GitHub repository[1]. The software is open source and contains comprehensive documentation with examples of use.

## 2  Related works

Packages for AutoML are prevalent in Python. The first AutoML solutions like Auto-WEKA (Thornton et al., 2013), was followed by Auto-Sklearn (Feurer et al., 2015, 2022) and TPOT (Tree-Based Pipeline Optimization Tool) (Olson et al., 2016) which was one of the very first AutoML methods and open-source software packages developed for the data science community in Python. But in R, there are few approaches. One of them is the *H2O* package (LeDell et al., 2022). It is an open-source library that is an in-memory, distributed, fast, and scalable machine learning and predictive analytics platform that creates a ranked list of models easily exported for use in a production environment. The authors have created an easy-to-use interface that automates the training of multiple candidate models. *H2O*'s AutoML is also designed for more advanced users by providing a simple wrapper function that performs many modeling tasks. *H2O*'s AutoML process automatically trains models and tunes them at user-specified times. To better understand the quality of models in *H2O*, we can rely on metrics such as $R^2$ and mean square error (MSE). For comparison, in the *forester* package, we can compare models using the most commonly used metrics or even define a new custom metric. What particularly distinguishes the *forester* package from *H2O* is the preprocessing. In the latter's case, it only includes target encoding and is in the experimental stage. In the *forester* package, we have more accurate and extensive preprocessing. In addition, *H2O* always requires Java to work, so the user must also install it.

The second widely-used framework is the *mlr3* package (Lang et al., 2019) which provides a framework for classification, regression, survival analysis, and other ML tasks such as cluster analysis. It provides the ability to perform hyperparameter tuning and feature selection. The package is well-documented, contains many functions and models, and provides many capabilities. However, it is different from a typical package for AutoML, as creating models requires knowledge of how to do it and some time to assemble such a model. It also has its drawbacks, such as the need for more preprocessing, which would help to use it more easily, for example, the XGBoost model, which has to have only numerical data without factors. There is also no way to divide the collection into training, testing, and validation subsets. The *mlr3* package provides functionality that builds on the basic components of machine learning. It can be extended to include preprocessing, pipelining, visualization, additional learners, additional task types, and more. To create these properties, we need to install many other libraries. In the *forester* package, we provide these components at once, and with a single function, we can perform preprocessing, prepare visualization of the results

---

[1]https://github.com/ModelOriented/forester

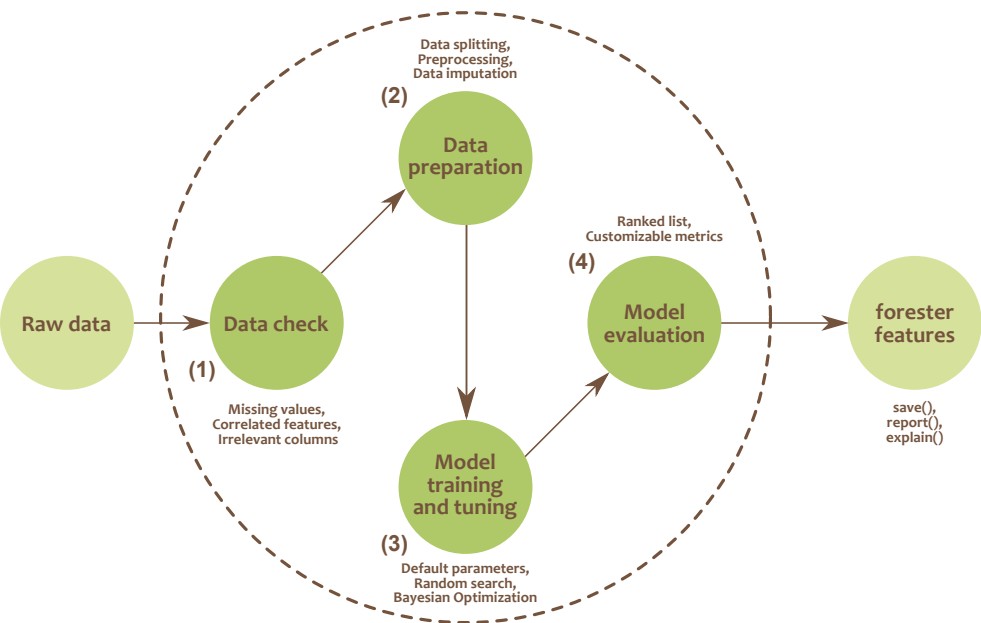

Figure 1: A diagram presenting the *forester* pipeline. The *forester* analyses poor-quality data with the in-built data check (1), which points to possible issues, and later data preparation (2) handles them during the preprocessing. In the next step, the models are trained with default and random searched parameters and tuned with a Bayesian optimization algorithm (3). In the end, trained models are evaluated (4) and presented as a ranked list. In addition, the package offers the user additional features.

and generate a report. A more detailed comparison of the *forester* package with *H2O* and *mlr3* is presented in Appendix F.

## 3 *forester* AutoML

The *forester* is an AutoML package automating the machine learning pipeline, starting from the data preparation, through model training, to the interpretability of the results. This way, we minimize the user's time performing basic and often repetitive activities related to the machine-learning process. Despite the high automation of the pipeline shown in Figure 1, we expose multiple parameters which advanced data scientists can use to customize the model creation. The whole package relies on the four pillars described in this section.

1. **Data check**
   The first one, called data check, concerns a data preparation phase. Data preparation is a crucial part of the modeling process (Rutkowski et al., 2010), so we cannot blindly assume a single way of transforming the data for all cases. Appropriate data preprocessing is crucial to building a model with a small error rate. To face that issue, we introduce a data check report summarizing the dataset with some basic information and pointing out possible problems. Data problems can affect the following modeling stages and be relevant to any model. The data check report points out id-like, duplicated, static, or highly correlated columns. Moreover, it points out the outliers, missing values, and the imbalance of the target. This way we can propose some simple heuristic data preprocessing methods, yet more advanced users are able to fight the issues mentioned by studying the data check report on their own.

2. **Data preparation**

   Preparing the data for modeling is another crucial aspect after checking the data. It can be done using a dedicated tool, but the *forester* package offers two general-purpose preprocessing methods, basic and advanced. The main purpose of this function is to remove the need to prepare data manually differently for different types of models. The basic preparation consists of the actions that are necessary for the package to work that is: the removal of static columns, binarization of the target variable, and imputation of the missing data using the MICE algorithm (Buuren and Groothuis-Oudshoorn, 2011). The advanced method additionally includes the removal of id-like columns (features suspected of being id), removal of highly correlated columns (Spearman's rank for the numerical features, and Crammer's V rank for categorical features) as well as feature selection with the BORUTA algorithm (Kursa and Rudnicki, 2010). Additionally, every model in the *forester* package requires a different data format which is also prepared inside the main function.

3. **Model training and tuning**

   The *forester* package's third and most important pillar is model training and tuning. Our solution focuses on the tree-based model family because of their high-quality performance for various tabular data tasks. We've limited ourselves to 5 well-known engines with different strong and weak points, so they complement each other.

   We have included the basic decision tree from partykit package (Hothorn and Zeileis, 2015) as an extremely light engine, but mostly, we have focused on the ensemble models. The only bagging representative is the random forest from the ranger package (Wright and Ziegler, 2017), which is reluctant to overfit.

   We have also considered three different boosting algorithms. The XGBoost model (Chen and Guestrin, 2016) is highly effective, but due to the need for one hot encoding, it suffers from the abundance of categorical features. However, the LightGBM model (Ke et al., 2017), which works best for medium and large datasets, has problems with the small ones. The last engine is the CatBoost (Prokhorenkova et al., 2018) which can achieve superior performance but requires the Java environment installed, which is a minor inconvenience.

   The models are trained with three approaches: using the default parameters, performing the random search algorithm within the predefined parameter space, and running an advanced Bayesian Optimization algorithm for fine-grained tuning. The first method is the baseline for other models. With the second one, we can cheaply create multiple models and explore various parameter combinations. The best and most time-consuming method is the Bayesian Optimization from the *ParBayesianOptimization* package. However, it is extremely useful for complex tasks.

4. **Model evaluation**

   The last pillar is the automatic evaluation of the trained models. The *forester* package assesses every trained model by various metrics, such as accuracy, area under the receiver operating characteristic curve (AUC), and F1 for the binary classification tasks, and Root Mean Squared Error (RMSE), Mean Absolute Error (MAE), or $R^2$ for the regression tasks. The results are later presented as a ranked list sorted by the outcomes (for example, ascending order for RMSE, and descending for AUC). Moreover, the user can define their metrics and provide them for the evaluation phase.

## 4 *forester* features

One of the most important goals for the *forester* package is the convenience of use and helping the users to focus more on analyzing the results instead of writing the code. To obtain such a user-friendly environment, the *forester* offers plenty of additional features useful for data scientists.

### 4.1 Model explanations

In recent years, interpretable machine learning has become a significant trend in machine learning. The tools providing interpretability such as *DALEX* (Biecek, 2018) or *iml* (Molnar et al., 2020) allow data scientists to explain how the models they create work, making it easier to detect their misbehavior. Models' explainability also enhances trust in such tools, even in demanding environments like medical researchers. To support using explainable methods for the models trained by the *forester*, we have created a wrapper for the DALEX explainer compatible with our package. This way, the user can easily create various explanations for the trained models.

### 4.2 Saving the outcomes

Another crucial feature is the save function, which lets the user save the training output. Returned *forester* object contains lots of information, such as preprocessed dataset, split datasets, split indexes, ranked lists for training, testing, and validation datasets, the predictions of the model, and much more. The abundance of objects makes it incredibly important to save the outcomes after the time-consuming training process.

### 4.3 Automated report

Last but not least, our solution offers an automatically generated report that helps users quickly and easily analyze the training results. The main goal of this feature is to ensure that every user is able to easily assess the quality of the trained models. The report consists of basic information about the dataset, a data check report, a ranked list of the best ten models, and visualizations concerning model quality. An example report for the *blood-transfusion-service-center* dataset (from the OpenML-CC18 benchmark (Bischl et al., 2021)) is provided in Appendix G.

The plots are divided into two groups; the first one compares the outcomes of different models, which helps to decide which model is the best. For example, guided by the radar chart comparison plot, we can choose the model with slightly worse accuracy, but better AUC and F1 values.

The second type of plots concentrates on the model with the best performance, and its most prominent feature is providing a feature importance plot. This visualization lets us understand which variables are the most important for the model; thus, we can evaluate its correctness. It is worth noticing that the reports, mostly visualizations, are different for binary classification and regression tasks as we measure their performance differently.

## 5 User interface

### 5.1 Training function

The *forester*'s main `train()` function runs the entire AutoML pipeline, including the data preparation, model training, and evaluation. To keep the package as simple as possible, the function requires only the dataset and target column name (Listing 1); however, to keep the tool versatile, there are lots of custom parameters for more advanced users (Listing 2). With the latter option, the user can specify the amount of Bayesian optimization iterations, the number of random search evaluations, proportions of the train, test, and validation subsets, change the preprocessing methods or even add their evaluation metric.

```
train_output ← train(data = lisbon, y = 'Price')
```
Listing 1: Training models with the *forester* package and default parameters.

```
train_output ← train(data = lisbon,
                      y = 'Price',
                      verbose = TRUE,
                      engine = c('ranger', 'xgboost', 'decision_tree',
                      'lightgbm', 'catboost'),
                      train_test_split = c(0.6, 0.2, 0.2),
                      bayes_iter = 10,
                      random_evals = 3,
                      advanced_preprocessing = FALSE,
                      metrics = 'auto',
                      sort_by = 'auto',
                      metric_function = NULL,
                      metric_function_name = NULL,
                      metric_function_decreasing = TRUE,
                      best_model_number = 5)
```
Listing 2: Training models with the *forester* package and custom parameters.

## 5.2 Extensive features

Apart from the `train()` function, the user can utilize additional functions, which is helpful during the modeling process. The `check_data()` function (Listing 3) enables printing a data check report outside of the `train()` function. The `save()` function (Listing 4) lets us save the outcome of the training process, whereas the `report()` function (Listing 5) creates a training report. The last extension is the `explain()` function (Listing 6), which creates a *DALEX* explainer that can be used to generate multiple visualizations concerning the model interpretability with the *DALEX* package.

```
check_data(data = `blood-transfusion-service-center`, y = 'Class')
```
Listing 3: Generating a data check report.

```
save(train_output, name = 'train_output.RData')
```
Listing 4: Saving the train output.

```
report(train_output, 'report.pdf')
```
Listing 5: Generating a report from the train output.

```
exp ← explain(models = train_output$best_models[[1]],
              test_data = train_output$data,
              y =  train_output$y,
              verbose = FALSE)
```
Listing 6: Creating a model explainer, that lets us use functions from the DALEX package.

## 6 Performance

To evaluate the performance of the package, we've decided to compare it to the *H2O* framework on the binary classification tasks from the OpenML-CC18 benchmark (Bischl et al., 2021) and regression tasks from OpenML (Vanschoren et al., 2013). Due to the limited computational resources, we have chosen a subset of 8 datasets for classification and 7 for regression described in Table 1 and Table 2, respectively. The binary classification datasets consisted mainly of categorical variables and contained many missing values, a significant obstacle for both solutions, whereas the regression tasks had no missing values and mostly numeric or binary values.

During the experiment, we trained the *forester* package three times for each dataset with random seeds provided for the data splitting function inside the *forester*. The same splits were later used for the *H2O* framework. A singular training iteration was executed for the decision tree, random forest, LightGBM, and CatBoost engines with ten iterations of the Bayesian optimization and ten random search evaluations. For the regression task we've additionally added an XGboost engine. To ensure that both frameworks had the same amount of time, we have measured it for every *forester* training iteration, and provided it to the respective *H2O* AutoML runs. This *H2O* functionality didn't work as supposed, and finally this framework had two times longer training time on average. This factor definitely improved the *H2Os* results, and we have to bear that in mind during the outcomes comparison. For further details see Appendix E. Additionally, to ensure the same data split, we have used the indexes saved during the *forester* training. The source codes are included in Appendix A.

The comparison of performance for both frameworks is presented in Figure 2 and Figure 3. For the raw results, as well as aggregated tabular ones, see Appendix C. As one can see, for the binary classification task, the *forester* outperformed the *H2O* framework on five datasets: *banknote-authentication*, *blood-transfusion-service-centre*, *credit-approval*, *credit-g*, and *diabetes*. The outcomes for very simple datasets *kr-vs-kp* and *breast-w* were similar, and *H2O* obtained better performance for the phoneme data. For the regression tasks, the results were comparable to the H2O's for most tasks or slightly worse, as for the *pol* dataset. The results show that the *forester* creates high-quality models that are competitive with the existing solutions.

However, our conclusions cannot be too far-fetched since we tested the package for only a few sets for binary classification and regression tasks. We cannot say that the *forester* package's predictive power is better than *H2O*, but they clearly are competitive.

Table 1: A subset of OpenML-CC18 benchmark datasets used during the evaluation process of the *forester* package, which are tabular data objects presenting the binary classification tasks. The features are mostly categorical, and they contain lots of missing values.

| Name | Number of columns | Number of rows |
|------|-------------------|----------------|
| kr-vs-kp | 37 | 3196 |
| breast-w | 10 | 699 |
| credit-approval | 16 | 690 |
| credit-g | 21 | 1000 |
| diabetes | 9 | 768 |
| phoneme | 6 | 5404 |
| banknote-authentication | 5 | 1372 |
| blood-transfusion-service-center | 5 | 748 |

Table 2: A subset of OpenML datasets used during the evaluation process of the *forester* package, which are tabular data objects presenting the regression tasks. In this case there were no missing values, and the features were mostly numerical or binary.

| Name | Number of columns | Number of rows |
|------|-------------------|----------------|
| bank32nh | 33 | 8192 |
| wine_quality | 12 | 6497 |
| Mercedes_Benz_Greener_Manufacturing | 378 | 4209 |
| kin8nm | 9 | 8192 |
| pol | 49 | 15000 |
| 2dplanes | 11 | 40768 |
| elevators | 19 | 16599 |

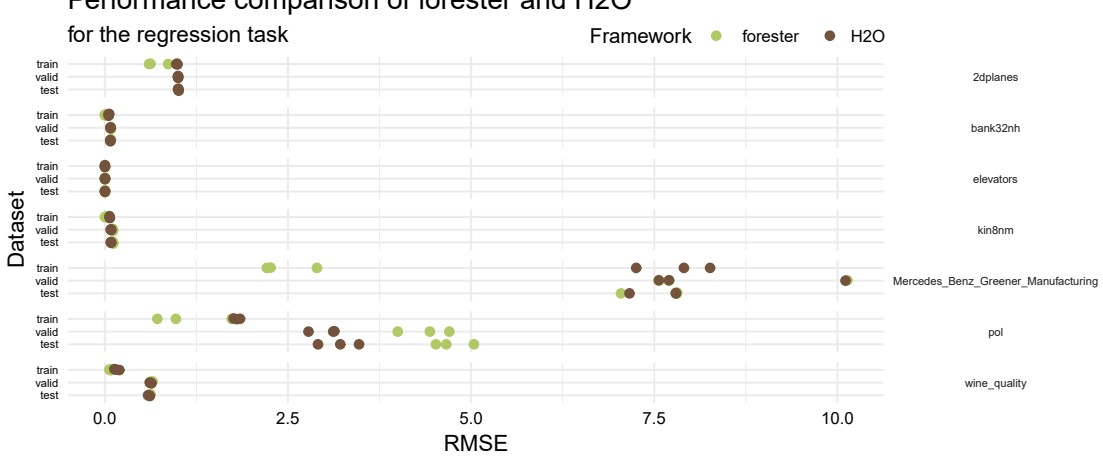

Figure 2: Performance comparison for *forester* and *H2O* frameworks for the datasets described in Table 1. Every experiment is conducted 3 times, which results in three observations visible on the plot for each dataset. Note that in some cases the dots might overlap. This plot clearly shows us that the *forester* performs better than the *H2O* package on the provided tasks, which confirms that it is a highly competitive framework.

Figure 3: Performance comparison for *forester* and *H2O* frameworks for the datasets described in Table 2. Every experiment is conducted 3 times, which results in three observations visible on the plot for each dataset. Note that in some cases the dots might overlap. This plot shows us that the *forester* performs comparably to the *H2O* package on the provided tasks, which confirms that it is a highly competitive framework.

## 7 Limitations and Broader Impact Statement

The *forester* package has limitations in the availability of models. The library contains only tree-based models, but this family proves to be extremely versatile. Only binary classification and regression are available in the current version of the package. Preparing models for multi-criteria classification, cluster analysis, or survival analysis is currently impossible. However, these features can be easily implemented in the future. The package currently performs better with smaller datasets; a large allocation of memory and time is needed for large and complex data.

One of the strongest points of the forester package is being incredibly easy to use, even if we do not have broad machine learning expertise. This approach, however, raises the risk that the models trained with the package will be of poor quality, for example, due to the training on a low-quality dataset, or that the outcomes will be misunderstood or incorrectly interpreted by the inexperienced user. The reporting module addresses all of these responsible machine learning concerns, which informs about possible issues with the data, measures the quality of the models, and provides their explanations.

## 8 Conclusions

This paper presents an R package for AutoML, creating models for regression and binary classification tasks conducted on tabular data. Our solution addresses the needs we have observed in AutoML tools in various programming languages. The main goals of the package are to keep the package stable and easy to use, to automate all the necessary steps inside the ML pipeline, and to provide results that are easy to create, understand and allow for diagnostics of the models. To achieve these results, we have focused only on the best representatives from the family of tree-based models that show superiority over other methods on tabular data. Furthermore, we provide additional functions that allow the user to save the models, create explanations and create a report describing the learning process and explaining the developed models. Experiments carried out tentatively indicate that more predictive power is obtained using our solution than currently existing solutions in R.

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

## A  Source Code

The source code of the experiments, prepared visualizations, and tables from Appendix C is available in the GitHub repository `https://github.com/ModelOriented/forester/tree/main/misc/experiments` as the `forester_benchmark.Rmd` file. The markdown notebook file describes the installation process, and it can be safely executed with the guidance of our remarks between the code chunks.

## B  Resources

As mentioned in the Section 6, our team was limited in computational power. The experiment was conducted on our private PC with 32GB of RAM, CPU: 11th Gen Intel(R) Core(TM) i7-11700KF @ 3.60GHz (16 cores), and the GPU: NVIDIA GeForce RTX 3070 Ti, however as the *forester* is not yet implemented to work on the GPU, only the CPU was used.

## C  Raw results

In this section we provide information about the raw results mentioned in the Section 6 which were used in the Figure 2. Raw results for train, test, and validation datasets are available in the GitHub repository `https://github.com/ModelOriented/forester/tree/main/misc/experiments/raw_training_results`. In this section we offer the results aggregated as the mean values of the metrics which are presented in the Table 3, Table 4, and Table 5 for the binary classification tasks. These tables also broaden our perspective by providing AUC and F1 values. The results for the regression tasks are presented in the Table 6, Table 7, and Table 8. These tables also broaden our perspective by providing MSE, R2, and MAE values.

Table 3: This table provides mean accuracy, AUC, and F1 values for the *forester* and *H2O* framework for all binary classification training datasets used in the benchmark.

| task_name | framework | accuracy | auc | f1 |
|---|---|---|---|---|
| banknote-authentication | forester | 1 | 1 | 1 |
| banknote-authentication | H2O | 0.929 | 0.923 | 0.905 |
| blood-transfusion-service-center | forester | 0.77 | 0.752 | 1 |
| blood-transfusion-service-center | H2O | 0.7 | 0.682 | 0.519 |
| breast-w | forester | 1 | 1 | 1 |
| breast-w | H2O | 0.998 | 0.998 | 0.997 |
| credit-approval | forester | 0.999 | 1 | 1 |
| credit-approval | H2O | 0.961 | 0.959 | 0.955 |
| credit-g | forester | 0.967 | 0.998 | 1 |
| credit-g | H2O | 0.906 | 0.855 | 0.938 |
| diabetes | forester | 0.991 | 0.999 | 1 |
| diabetes | H2O | 0.874 | 0.871 | 0.826 |
| kr-vs-kp | forester | 1 | 1 | 1 |
| kr-vs-kp | H2O | 0.999 | 0.999 | 0.965 |
| phoneme | forester | 1 | 1 | 1 |
| phoneme | H2O | 1 | 1 | 1 |

Table 4: This table provides mean accuracy, AUC, and F1 values for the *forester* and *H2O* framework for all binary classification testing datasets used in the benchmark.

| task_name | framework | accuracy | auc | f1 |
|---|---|---|---|---|
| banknote-authentication | forester | 0.995 | 0.995 | 1 |
| banknote-authentication | H2O | 0.933 | 0.927 | 0.915 |
| blood-transfusion-service-center | forester | 0.796 | 0.772 | 0.976 |
| blood-transfusion-service-center | H2O | 0.713 | 0.707 | 0.54 |
| breast-w | forester | 0.976 | 0.984 | 0.986 |
| breast-w | H2O | 0.971 | 0.97 | 0.959 |
| credit-approval | forester | 0.885 | 0.931 | 0.942 |
| credit-approval | H2O | 0.882 | 0.882 | 0.87 |
| credit-g | forester | 0.733 | 0.79 | 0.865 |
| credit-g | H2O | 0.743 | 0.64 | 0.829 |
| diabetes | forester | 0.768 | 0.823 | 0.799 |
| diabetes | H2O | 0.753 | 0.727 | 0.643 |
| kr-vs-kp | forester | 0.994 | 0.999 | 0.991 |
| kr-vs-kp | H2O | 0.991 | 0.991 | 0.991 |
| phoneme | forester | 0.909 | 0.96 | 0.867 |
| phoneme | H2O | 0.904 | 0.895 | 0.842 |

Table 5: This table provides mean accuracy, AUC, and F1 values for the *forester* and *H2O* framework for all binary classification validation datasets used in the benchmark.

| task_name | framework | accuracy | auc | f1 |
|---|---|---|---|---|
| banknote-authentication | forester | 1 | 1 | 1 |
| banknote-authentication | H2O | 0.916 | 0.908 | 0.887 |
| blood-transfusion-service-center | forester | 0.775 | 0.773 | 0.833 |
| blood-transfusion-service-center | H2O | 0.675 | 0.68 | 0.509 |
| breast-w | forester | 0.938 | 0.968 | 0.956 |
| breast-w | H2O | 0.967 | 0.97 | 0.953 |
| credit-approval | forester | 0.855 | 0.908 | 0.939 |
| credit-approval | H2O | 0.867 | 0.862 | 0.842 |
| credit-g | forester | 0.705 | 0.788 | 1 |
| credit-g | H2O | 0.758 | 0.635 | 0.846 |
| diabetes | forester | 0.747 | 0.803 | 0.866 |
| diabetes | H2O | 0.755 | 0.735 | 0.656 |
| kr-vs-kp | forester | 0.99 | 0.999 | 0.99 |
| kr-vs-kp | H2O | 0.99 | 0.99 | 0.99 |
| phoneme | forester | 0.901 | 0.954 | 0.851 |
| phoneme | H2O | 0.9 | 0.896 | 0.839 |

Table 6: This table provides mean RMSE, MSE, $R^2$, and MAE values for the *forester* and *H2O* framework for all regression training datasets used in the benchmark.

| task_name | framework | rmse | mse | r2 | mae |
|---|---|---|---|---|---|
| 2dplanes | forester | 0.697 | 0.5 | 0.974 | 0.423 |
| 2dplanes | H2O | 0.984 | 0.969 | 0.95 | 0.785 |
| bank32nh | forester | 0.001 | 0 | 1 | 0.001 |
| bank32nh | H2O | 0.054 | 0.003 | 0.806 | 0.037 |
| elevators | forester | 0.001 | 0 | 0.978 | 0.001 |
| elevators | H2O | 0.002 | 0 | 0.942 | 0.001 |
| kin8nm | forester | 0.012 | 0 | 0.997 | 0.009 |
| kin8nm | H2O | 0.066 | 0.004 | 0.937 | 0.051 |
| Mercedes_Benz_Greener_Manufacturing | forester | 2.456 | 6.13 | 0.963 | 0.775 |
| Mercedes_Benz_Greener_Manufacturing | H2O | 7.806 | 61.115 | 0.625 | 4.935 |
| pol | forester | 1.139 | 1.483 | 0.999 | 0.699 |
| pol | H2O | 1.803 | 3.251 | 0.998 | 0.829 |
| wine_quality | forester | 0.071 | 0.005 | 0.993 | 0.031 |
| wine_quality | H2O | 0.161 | 0.027 | 0.965 | 0.124 |

Table 7: This table provides mean RMSE, MSE, $R^2$, and MAE values for the *forester* and *H2O* framework for all regression testing datasets used in the benchmark.

| task_name | framework | rmse | mse | r2 | mae |
|---|---|---|---|---|---|
| 2dplanes | forester | 1.003 | 1.007 | 0.948 | 0.802 |
| 2dplanes | H2O | 1.004 | 1.008 | 0.948 | 0.802 |
| bank32nh | forester | 0.08 | 0.006 | 0.548 | 0.053 |
| bank32nh | H2O | 0.076 | 0.006 | 0.599 | 0.05 |
| elevators | forester | 0.002 | 0 | 0.884 | 0.002 |
| elevators | H2O | 0.002 | 0 | 0.911 | 0.001 |
| kin8nm | forester | 0.113 | 0.013 | 0.816 | 0.087 |
| kin8nm | H2O | 0.084 | 0.007 | 0.899 | 0.065 |
| Mercedes_Benz_Greener_Manufacturing | forester | 7.554 | 57.195 | 0.626 | 5.039 |
| Mercedes_Benz_Greener_Manufacturing | H2O | 7.583 | 57.598 | 0.623 | 5.222 |
| pol | forester | 4.739 | 22.508 | 0.987 | 2.242 |
| pol | H2O | 3.198 | 10.278 | 0.994 | 1.3 |
| wine_quality | forester | 0.614 | 0.377 | 0.505 | 0.451 |
| wine_quality | H2O | 0.604 | 0.365 | 0.521 | 0.43 |

Table 8: This table provides mean RMSE, MSE, $R^2$, and MAE values for the *forester* and *H2O* framework for all regression validation datasets used in the benchmark.

| task_name | framework | rmse | mse | r2 | mae |
|---|---|---|---|---|---|
| 2dplanes | forester | 0.999 | 0.997 | 0.948 | 0.799 |
| 2dplanes | H2O | 1 | 0.999 | 0.948 | 0.8 |
| bank32nh | forester | 0.082 | 0.007 | 0.544 | 0.053 |
| bank32nh | H2O | 0.078 | 0.006 | 0.591 | 0.052 |
| elevators | forester | 0.002 | 0 | 0.875 | 0.002 |
| elevators | H2O | 0.002 | 0 | 0.907 | 0.001 |
| kin8nm | forester | 0.111 | 0.012 | 0.822 | 0.085 |
| kin8nm | H2O | 0.083 | 0.007 | 0.899 | 0.065 |
| Mercedes_Benz_Greener_Manufacturing | forester | 8.464 | 73.039 | 0.559 | 5.261 |
| Mercedes_Benz_Greener_Manufacturing | H2O | 8.458 | 72.911 | 0.56 | 5.373 |
| pol | forester | 4.379 | 19.256 | 0.989 | 1.885 |
| pol | H2O | 3.01 | 9.087 | 0.995 | 1.213 |
| wine_quality | forester | 0.632 | 0.399 | 0.478 | 0.466 |
| wine_quality | H2O | 0.624 | 0.389 | 0.492 | 0.447 |

## D Used assets

In this section we describe the packages used for both *forester*, and the experiments. The packages outside of the *forester* required for the experiments are listed in the Table 9. Additional requirement for the *catboost* and *H2O* packages is installed Java. The packages required by the *forester* as well as their versions used during the experiment are presented in the Table 10.

Table 9: The packages and their versions under which the experiments were executed and supplemental materials were created.

| package | version | license |
|---|---|---|
| xlsx | 0.6.5 | GPL-3 |
| stringr | 1.5.0 | MIT |
| ggbeeswarm | 0.6.0 | GPL (>= 2) |
| dplyr | 1.0.10 | MIT |
| ggplot2 | 3.4.0 | MIT |
| tictoc | 1.1 | Apache License (== 2.0) |
| H2O | 3.38.0.1 | Apache License (== 2.0) |
| forester | 1.2.1 | GPL-3 |
| OpenML | 1.12 | BSD_3_clause |

Table 10: The *forester* package's dependencies and their versions used during the experiments.

| package | version | licence |
|---|---|---|
| Boruta | 7.0.0 | GPL (>= 2) |
| catboost | 1.1.1 | Apache License (== 2.0) |
| crayon | 1.5.2 | MIT |
| DALEX | 2.4.2 | GPL |
| data.table | 1.14.2 | MPL-2.0 |
| ggplot2 | 3.4.0 | MIT |
| ggradar | 0.2 | GPL |
| ggrepel | 0.9.3 | GPL-3 |
| knitr | 1.40 | GPL |
| lightgbm | 3.3.2 | MIT |
| mice | 3.14.0 | GPL-2 | GPL-3 |
| mltools | 0.3.5 | MIT |
| ParBayesianOptimization | 1.2.4 | GPL-2 |
| partykit | 1.2-16 | GPL-2 | GPL-3 |
| pROC | 1.18.0 | GPL (>= 3) |
| ranger | 0.14.1 | GPL-3 |
| rcompanion | 2.4.18 | GPL-3 |
| rmarkdown | 2.16 | GPL-3 |
| splitTools | 0.3.2 | GPL (>= 2) |
| testthat | 3.1.6 | MIT |
| tibble | 3.1.8 | MIT |
| tinytex | 0.43 | MIT |
| varhandle | 2.0.5 | GPL (>= 2) |
| xgboost | 1.6.0.1 | Apache License (== 2.0) |
| stats | 4.1.2 | Part of R 4.1.2 |

## E Execution times comparison

In this section we briefly explore the times needed for every experiment execution for both frameworks. The results presented in Table 11, and Table 12 show that final execution times differ, despite setting exactly the same times for *H2O* experiment as the forester had. Our empirical results show that the *H2O* runs lasted two times longer on average than the *forester*, which puts a different light on the comparison of the frameworks performance. Raw results needed for these tables are available in the GitHub repository `https://github.com/ModelOriented/forester/tree/main/misc/experiments/execution_times`.

Table 11: The comparison of mean execution times in seconds for the *forester* and *H2O* for binary classification experiments.

| task_name | forester | H2O | difference | relative difference |
|---|---|---|---|---|
| banknote-authentication | 818.33 | 2521.33 | -1703 | 0.28 |
| blood-transfusion-service-center | 155.67 | 555.67 | -400 | 0.26 |
| breast-w | 451.33 | 797.33 | -346 | 0.57 |
| credit-approval | 805 | 1513 | -708 | 0.53 |
| credit-g | 2453 | 4234 | -1781 | 0.58 |
| diabetes | 1645.67 | 2643.67 | -998 | 0.62 |
| kr-vs-kp | 451.33 | 806.67 | -355.33 | 0.57 |
| phoneme | 2748.33 | 3695.33 | -947 | 0.67 |

Table 12: The comparison of mean execution times in seconds for the *forester* and *H2O* for regression experiments.

| task_name | forester | H2O | difference | relative difference |
|---|---|---|---|---|
| 2dplanes | 401 | 1050.67 | -649.67 | 0.38 |
| bank32nh | 708.67 | 1214.67 | -506 | 0.58 |
| elevators | 720.33 | 1435.33 | -715 | 0.5 |
| kin8nm | 544.67 | 1564 | -1019.33 | 0.35 |
| Mercedes_Benz_Greener_Manufacturing | 848 | 1371.67 | -523.67 | 0.61 |
| pol | 756 | 1548.33 | -792.33 | 0.49 |
| wine_quality | 1317.33 | 2130 | -812.67 | 0.63 |

## F Package comparison

We have prepared a notebook showing the differences between the packages described in the related work section. The document includes a comparison of package installation, a description of available preprocessing, variable selection options, and model tuning. In addition, visualizations, methods of explainable machine learning, report preparation, and reference to available package documentation are described. We do not give a final assessment of the best package because it could be subjective, but we expose the reader to criticism. Notebook is available in the GitHub repository `https://github.com/ModelOriented/forester/blob/main/misc/experiments/framework_comparison.Rmd`.

## G Report example

# Forester report

version 1.2.1

2023-05-20 01:36:36

This report contains details about the best trained model, table with metrics for every trained model, scatter plot for chosen metric and info about used data.

## The best models

This is the **binary__clf** task.
The best model is: **xgboost__RS__5**.

The names of the models were created by a pattern *Engine__TuningMethod__Id*, where:

- Engine describes the engine used for the training (random_forest, xgboost, decision_tree, lightgbm, catboost),

- TuningMethod describes how the model was tuned (basic for basic parameters, RS for random search, bayes for Bayesian optimization),

- Id for separating the random search parameters sets.

*More details about the best model are present at the end of the report.*

| no. | name | accuracy | auc | f1 |
|----:|------|---------:|-------:|-------:|
| 13 | xgboost__RS__5 | 0.7919 | 0.8088 | 0.2791 |
| 7 | ranger__RS__4 | 0.7785 | 0.6965 | 0.1538 |
| 18 | lightgbm__RS__5 | 0.7785 | 0.7361 | 0.4211 |
| 2 | xgboost__model | 0.7718 | 0.7090 | 0.4138 |
| 14 | lightgbm__RS__1 | 0.7718 | 0.7578 | 0.3704 |
| 4 | ranger__RS__1 | 0.7651 | 0.7930 | NaN |
| 6 | ranger__RS__3 | 0.7651 | 0.7228 | NaN |
| 10 | xgboost__RS__2 | 0.7651 | 0.7801 | NaN |
| 11 | xgboost__RS__3 | 0.7651 | 0.7367 | NaN |
| 16 | lightgbm__RS__3 | 0.7651 | 0.7690 | NaN |
| 21 | lightgbm__bayes | 0.7651 | 0.7340 | 0.3636 |
| 8 | ranger__RS__5 | 0.7584 | 0.7579 | 0.0526 |
| 12 | xgboost__RS__4 | 0.7517 | 0.6609 | 0.3729 |
| 19 | ranger__bayes | 0.7517 | 0.7333 | 0.2449 |
| 20 | xgboost__bayes | 0.7517 | 0.7409 | 0.2449 |
| 1 | ranger__model | 0.7450 | 0.7063 | 0.3214 |
| 3 | lightgbm__model | 0.7450 | 0.6842 | 0.3871 |
| 9 | xgboost__RS__1 | 0.7450 | 0.6619 | 0.3667 |
| 15 | lightgbm__RS__2 | 0.7181 | 0.6058 | 0.3824 |
| 17 | lightgbm__RS__4 | 0.7181 | 0.6058 | 0.3824 |

| no. | name | accuracy | auc | f1 |
|---|---|---|---|---|
| 5 | ranger__RS__2 | 0.7114 | 0.6929 | 0.2712 |

**Plots for all models**

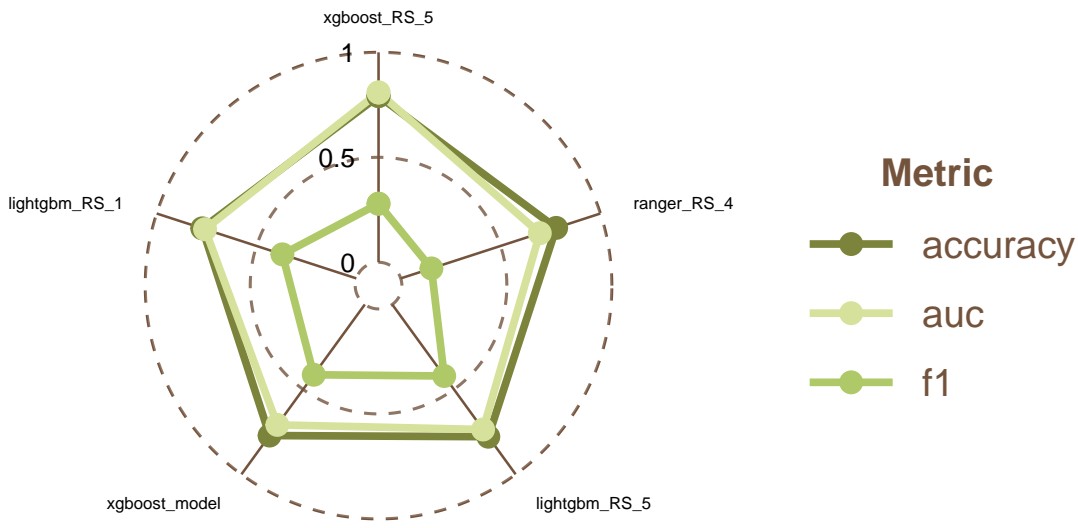

**Plots for the best model - xgboost__RS__5**

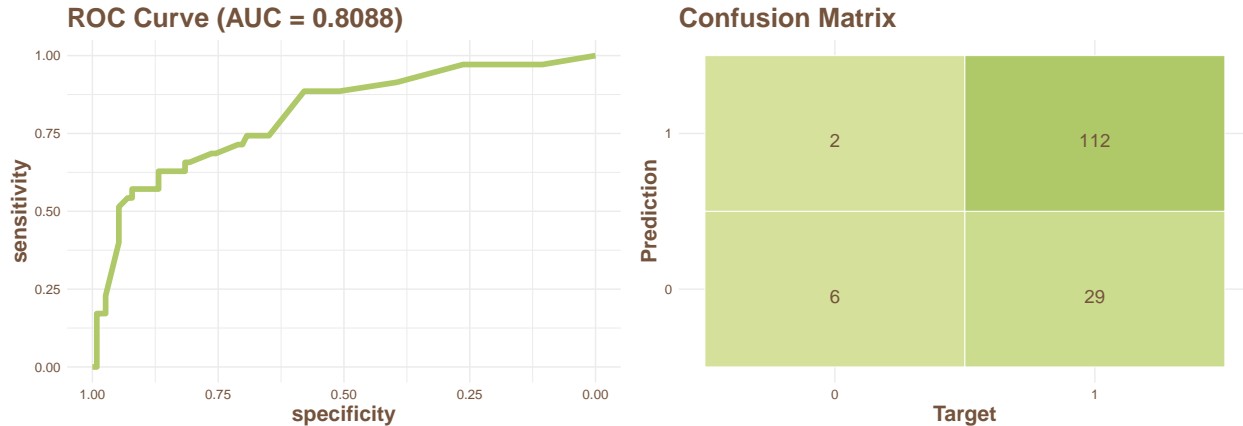

**Feature Importance for the best model - xgboost__RS__5**

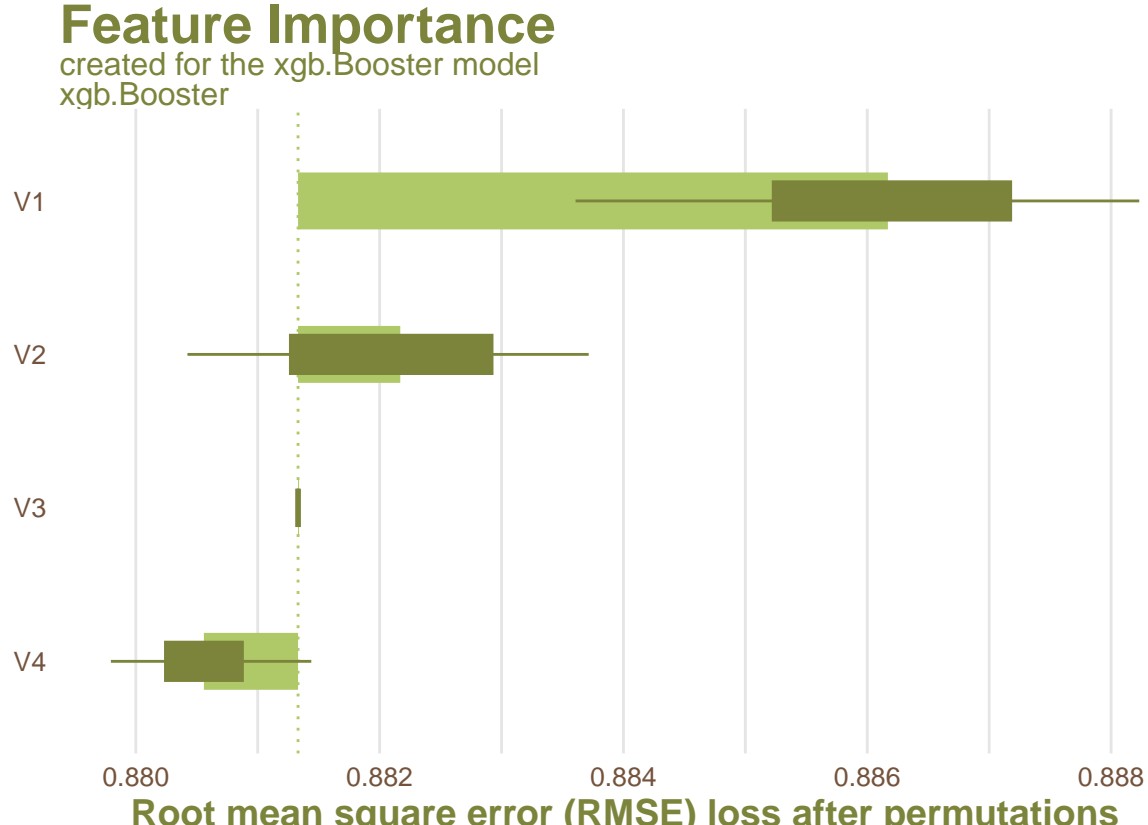

# Feature Importance
created for the xgb.Booster model
xgb.Booster

**Root mean square error (RMSE) loss after permutations**

## Details about data

──────────── **CHECK DATA REPORT** ────────────

**The dataset has 748 observations and 5 columns which names are:**

V1; V2; V3; V4; Class;

**With the target value described by a column:** Class.

**No static columns.**

**No duplicate columns.**

**No target values are missing.**

**No predictor values are missing.**

**No issues with dimensionality.**

**Strongly correlated, by Spearman rank, pairs of numerical values are:**

V2 - V3: 1;

**These observations migth be outliers due to their numerical columns values:**

1 10 116 342 496 497 498 499 5 500 501 503 504 505 506 518 529 747 748 ;

**Dataset is unbalanced with:** 3.202247 proportion with 1 being a dominating class.

Columns names suggest that none of them are IDs.

Columns data suggest that none of them are IDs.

——————— **CHECK DATA REPORT END** ———————

# The best model details

```
------------ Xgboost model ------------

Parameters
  niter: 20
  evaluation_log:
    iter : train_auc
      1 :
      2 :
      3 :
      4 :
      5 :
      6 :
      7 :
      8 :
      9 :
     10 :
     11 :
     12 :
     13 :
     14 :
     15 :
     16 :
     17 :
     18 :
     19 :
     20 :
```

