# OpenReview forum: "forester: A Novel Approach to Accessible and Interpretable AutoML for Tree-Based Modeling"
_automl.cc/AutoML/2023/ABCD_Track — AutoML 2023 (ABCD Track) asistoworkshop_

### Official Review · Reviewer_yWmt · 2023-05-09

**Potential Impact On The Field Of Automl Rating:** 3
**Technical Quality And Correctness:** The quality of the code is good and t…
**Technical Quality And Correctness Rating:** 4
**Clarity Rating:** 3

**Summary Of Contributions:**

This work introduces the forester package in the R language The package provides an easy-to-use framework for beginners and experts alike to use for AutoML using tree-based methods on tabular data.

The presentation includes discussion on the similarities and differences to two existing packages. It presents the package's main components of its AutoML pipeline as well as some additional features for saving and understanding the results.

It also includes a set of benchmarking results in classification, showing forester comparing well to another existing package.

**Actions Required To Increase Overall Recommendation:**

Actions that would increase my rating are:
* Please clarify the similarities and differences to other existing packages (especially tidymodels and caret, in addition to H2O and mlr3 which are discussed). A list or table of features available in each would be helpful.
* Results for regression benchmarking.
* Results for how fast forester is compared to e.g. H2O or another existing package.

**Clarity:**

The paper is well-written and easy to read. There are a few typos or unclear grammar at times, but overall the quality is good.

**Overall Review:**

### Strengths
* The proposed package covers the full AutoML pipeline, not just optimisation.
* Provides an easy-to-use framework that can be especially useful for non-experts.
* It introduces a novel and useful data quality checking step.
* It allows for HPO using Bayesian optimisation, random search or setting default params.
* Uses the DALEX package for explaining model behaviour.
* The code repository seems popular, with 93 stars, good issue handling, and a history of several developers contributing.
* It is an open-source software package under the GPL 3 license.
* This package will allow applying AutoML to real-world applications in the R language.
* The paper is well written.

### Weaknesses
* Provides similar functionality to the tidymodels, caret, mlr3 and H2O packages, but adds extra features like data quality checking.
* It does not support multinomial classification.
* The experimental evaluation is only performed for classification, not for regression.
* There is no evaluation of the time complexity of the package, i.e. how fast it is compared to H2O when performing the same task.

**Potential Impact On The Field Of Automl:**

This package can help bring the AutoML pipeline to a new community using the R language. It will likely be used for both practical use in industry as well as future research.

I am a bit unclear, however, on how novel this package is compared to other existing frameworks like tidymodels, caret, H2O and ml3.

**Review Confidence:**

4: You are confident in your assessment, but not absolutely certain. It is unlikely, but not impossible, that you did not understand some parts of the submission or that you are unfamiliar with some pieces of related work.

**Review Rating:**

6: Borderline Leaning Accept: Technically sound submission where reasons to accept outweigh reasons to reject. Please use sparingly.

**Review Summary:**

A novel framework that can be very useful for R users who are new to the AutoML process. I lean towards accepting the paper but am concerned about the package's similarities to existing frameworks as well as missing experimental evidence.

---

### Official Review · Reviewer_8q34 · 2023-05-10

**Potential Impact On The Field Of Automl Rating:** 1
**Technical Quality And Correctness Rating:** 1
**Clarity Rating:** 3

**Summary Of Contributions:**

The authors propose forester, an open source AutoML system written in R that performs data checks, data preprocessing, model training, and model evaluation in an easy to use interface. forester uses tree models including GBMs, along with hyperparameter tuning. A benchmark on 8 small datasets shows that forester obtains similar performance to H2O AutoML, winning on 5 out of 8 datasets.

**Actions Required To Increase Overall Recommendation:**

I encourage the authors to fix the issues mentioned regarding their experiments, and expand the experiments significantly to cover many more datasets. It would benefit the relevance of the paper greatly if it were to integrate into the AutoMLBenchmark for example (https://github.com/openml/automlbenchmark).

The authors should additionally perform ablations on their key contributions to enhance their impact and trustworthiness.

While addressing the above would enhance my score, I struggle to see relevant impact potential for the work that justifies acceptance unless from a performance standpoint it compares favorably to existing systems or the ablations reveal compelling and novel results.

**Clarity:**

The paper is generally well written and clear, however the choice of datasets for the benchmark is not well specified, and the reasoning for the very limited experiments is not well justified beyond claims of lacking compute resources (for example, no information on training time was given, nor mention of compute usage used to conduct the experiments).

**Overall Review:**

While forester does have positive aspects such as its simple design, automated data preprocessing, friendly API, easy to understand result reporting, and open-source availability, these aspects are in general available in numerous widely adopted, powerful, and easy to use open-source AutoML frameworks (ex: H2O AutoML, Auto-sklearn, FLAML, AutoGluon). The benefit of being available in R is an interesting aspect, however H2O AutoML is also available in R, and the experiments conducted are insufficient to meaningfully compare the systems. Further, availability in R is questionable to its impact, and the repository on GitHub lacks indication of significant user adoption compared to the aforementioned Python-based AutoML systems. The main positive aspect in my mind is the data cleaning, preprocessing, and result reporting, however these aspects are not explored via ablation studies, and are not sufficient in and of themselves to make a compelling AutoML system. Beyond the above, the experiments conducted appear to have correctness issues as I describe in detail earlier in the review.

**Potential Impact On The Field Of Automl:**

Because most of the base components within forester have already been implemented across many prior and actively developed AutoML systems, particularly in Python, the impact of the paper hinges on the relevancy of providing an R-based AutoML system, the utility provided by the data cleaning and preprocessing logic, and/or the performance of the AutoML system as a whole. However, the paper does not dive into sufficient detail on its advantages over the R offering of H2O AutoML, and the benchmark is too small to be used as an indicator.

The remaining point is regarding the data preprocessing, however this preprocessing is inherently tied to the forester system, and thus is only relevant when considering how it impacts the final model performance, which was not evaluated, as there are no ablation experiments. The final remaining point is the data cleaning, such as the removal of id-like columns and correlated features. However, these data cleaning steps have not been ablated within the work, and it is unclear what impact they have on the results. Because of the lack of depth in evaluation, I do not see much impact that can be gleaned from the paper.

**Review Confidence:**

5: You are absolutely certain about your assessment. You are very familiar with the related work and checked all the details carefully.

**Review Rating:**

3: Reject: For instance, a submission with technical flaws and limited impact.

**Review Summary:**

I am inclined to reject the paper due to the lack of meaningful experiments, ablation studies, and baselines, as well as the questionable utility forester provides beyond existing AutoML systems. Finally, the experiments that were conducted appear to have major correctness issues.

**Technical Quality And Correctness:**

I have no issue with the general description of the design and features of forester. The design is sensible and a viable AutoML implementation.

My primary issue is that the benchmark used is far too small to warrant any meaningful takeaways, and while I can understand that compute can be hard to come by, the datasets tested on are tiny (<6000 rows, <40 columns, binary classification).

For tree models such as Random Forest, LightGBM or XGBoost, they should be able to train on these datasets in seconds (or less), even on old laptops. Therefore, I struggle to understand how the authors were not able to include more datasets, include baselines, perform cross-validation, or conduct ablation studies. The lack of any larger dataset raises major concerns on the viability of forester beyond toy problems, and the lack of training time information in the results leaves it ambiguous if the training times are reasonable given the small size of the datasets.

Beyond these issues, I see severe problems in the actual benchmark results presented in Figure 2 and Tables 2, 3, & 4:

1. In Table 2, blood-transfusion-service-center forester has an accuracy of 1.0, but auc of 0.77 and f1 of 0.752. This is impossible. An accuracy of 1.0 requires an AUC and F1 score of 1.0, so something is being computed incorrectly. Beyond this example, there are various others where similar oddities are occurring (val credit-g, test blood-transfusion, etc.), but would be too lengthy to write out verbatim.

You can see this strange scoring issue in Appendix E as well, with `ranger_RS_3` having an accuracy of 1.0, but somehow having an auc of 0.77 and an f1 of 0.73, which again, should be impossible. Additionally, `catboost_RS_4` has an accuracy of 0.42 (aka worse than random guess), but an auc of 0.726 and an f1 of 0.7057. This severe issue is seemingly overlooked, yet is core to the entire verification of the results and utility as an AutoML system to practitioners.

The same strangeness occurs in Appendix E "Plots for all models", where the ranger models somehow have near perfect AUC while the other models have less than 0.5 AUC.

---

### Official Review · Reviewer_W7EP · 2023-05-10

**Potential Impact On The Field Of Automl:** 1. The proposed "forester" package ad…
**Potential Impact On The Field Of Automl Rating:** 3
**Technical Quality And Correctness Rating:** 2
**Clarity Rating:** 4

**Summary Of Contributions:**

1. The paper presents an open-source AutoML package called "forester" implemented in R language.
2. The package is designed to train tree-based models on tabular data for regression and binary classification tasks.
3. The motivation for developing this package is to address the limited R solutions available for AutoML, which have high entry levels and are not accessible to everyone. The "forester" package is easy to use and accessible to users with varying degrees of ML proficiency.
4. It automates various steps inside the ML pipeline, such as handling unprocessed datasets, feature engineering, and hyperparameter tuning using Bayesian optimization.
5. It provides additional functions allowing the user to save the models, create explanations, and a detailed report describing the learning process and the developed models. The report includes the ranked list of models, a comparison of models, and explanations for the best one.
6. Furthermore, experimentation with the "forester" package indicates that it can create models that are competitive with the existing solutions.

**Actions Required To Increase Overall Recommendation:**

Authors can take the following actions:
1. The authors have tested the package for the binary classification tasks and left the regression tasks without a thorough evaluation. It would be beneficial if they could also provide a detailed evaluation of the regression tasks.
2. For the performance evaluation, the authors have only compared the "forester" package to the H2O framework, which may not be comprehensive enough and representative of all available AutoML solutions. It would be helpful if they could provide more detailed comparisons with other AutoML solutions to demonstrate how their approach compares in terms of performance and ease of use.
3. Although they have briefly discussed the potential limitations and drawbacks towards the end of the paper, the paper could benefit from more detailed insights into the limitations of their approach and potential future work to address these limitations, such as the package's applicability to other types of data or tasks.

**Clarity:**

The paper is well-written, well-structured, and easy to understand, with clear and concise explanations of the ideas, methods used, and their implementation in the "forester" package.

Minor Correction:
There is one typo in the paper on Page 7. It should be "blood-transfusion-service-center" instead of "blood-transfusion-service-centre".

**Overall Review:**

The "forester" package presented in the paper offers a useful contribution to the field of AutoML, especially for those who prefer to work with the R language. The package is designed to train tree-based models on tabular data for regression and binary classification tasks. It is easy to use and accessible to users with varying degrees of ML proficiency, which can save both time and effort. It automates various steps inside the ML pipeline, such as handling unprocessed datasets, feature engineering, and hyperparameter tuning using Bayesian optimization. It provides additional functions allowing the user to save the models, create explanations, and a detailed report describing the learning process and the developed models. To evaluate the performance of the package, the authors have compared it to the H2O framework on binary classification tasks from the OpenML-CC18 benchmark. The results show that the "forester" creates models that are competitive with the existing solutions.

Areas to further improve the paper:
1. The authors have tested the package for the binary classification tasks and left the regression tasks without a thorough evaluation. It would be beneficial if they could also provide a detailed evaluation of the regression tasks.
2. For the performance evaluation, the authors have only compared the "forester" package to the H2O framework, which may not be comprehensive enough and representative of all available AutoML solutions. It would be helpful if they could provide more detailed comparisons with other AutoML solutions to demonstrate how their approach compares in terms of performance and ease of use.
3. Although they have briefly discussed the potential limitations and drawbacks towards the end of the paper, the paper could benefit from more detailed insights into the limitations of their approach and potential future work to address these limitations, such as the package's applicability to other types of data or tasks.

**Review Confidence:**

3: You are fairly confident in your assessment. It is possible that you did not understand some parts of the submission or that you are unfamiliar with some pieces of related work.

**Review Rating:**

7: Weak Accept: Technically sound paper with moderate-to-high impact, with perhaps some minor flaws.

**Review Summary:**

The paper presents a useful contribution to the field of AutoML, especially for those who prefer to work with the R language. The package is designed to train tree-based models on tabular data for regression and binary classification tasks. It is easy to use and accessible to users with varying degrees of ML proficiency. The package automates various steps in the ML pipeline and provides additional functions allowing the user to save models, create explanations, and a detailed training report. However, the paper could benefit from some improvements, such as providing detailed evaluation results of the regression tasks, comprehensive comparisons with more AutoML solutions besides the H2O framework, and a more detailed discussion of the potential limitations and drawbacks. Thus, I recommend a weak acceptance of this paper for the AutoML 2023 ABCD Track.

**Technical Quality And Correctness:**

The "forester" package is an open-source software package with a GPL-3 license. It can be found on GitHub - https://github.com/ModelOriented/forester. The GitHub repository currently has an active commit history, an active issue tracker, and 93 stars.

To evaluate the performance of the package, the authors have compared it to the H2O framework on binary classification tasks from the OpenML-CC18 benchmark. It outperformed the H2O framework on five datasets: banknote-authentication, blood-transfusion-service-center, credit-approval, credit-g, and diabetes. The outcomes for very simple datasets kr-vs-kp and breast-w were similar, and H2O obtained better performance for the phoneme data. The results show that the "forester" creates models that are competitive with the existing solutions.

Areas to further improve the technical quality of this paper:
1. The authors have tested the package for the binary classification tasks and left the regression tasks without a thorough evaluation. The paper would benefit if they could also provide a detailed evaluation of regression tasks.
2. For the performance evaluation, the authors have only compared the "forester" package to the H2O framework, which may not be comprehensive enough and representative of all available AutoML solutions. It would be helpful if they could provide more detailed comparisons with other AutoML solutions to demonstrate how their approach compares in terms of performance and ease of use.
3. Although they have briefly discussed the potential limitations and drawbacks towards the end of the paper, the paper could benefit from more detailed insights into the limitations of their approach and potential future work to address these limitations, such as the package's applicability to other types of data or tasks.

---

### Official Review · Reviewer_rS7P · 2023-05-11

**Potential Impact On The Field Of Automl Rating:** 3
**Technical Quality And Correctness Rating:** 3
**Clarity Rating:** 3
**Actions Required To Increase Overall Recommendation:** Address the issues I mention in my pr…

**Summary Of Contributions:**

The paper presents an R package "forester",  an AutoML package for training tree-based models on tabular data. The current implementation supports regression and binary classification.

**Details Of Ethical Concerns (Optional):**

/

**Clarity:**

The paper is well-written and structured.
More clarity should be provided by adding arguments for selecting the methods that are involved in each step.

**Overall Review:**

Positive: Nice work, especially focusing on AutoML in R programming language. This is an important step for bringing AutoML to groups that are more familiar with biostatistics and now involving ML in their experiments.

Negative: More evaluation should be performed (more benchmarks involved), especially the regression task since the package has been reported also for solving regression tasks. Can you also estimate the impact of each step and the method selected there on the final results obtained by the pipeline, just to estimate the contribution of each step in the modeling process?

**Potential Impact On The Field Of Automl:**

The impact is more on bringing AutoML to the data science community familiar with doing experiments in the R programming language. This is especially important to groups that are more related to biostatistics and interested to involve ML in their new studies.

**Reproducibility (Optional):**

I have not checked it.

**Review Confidence:**

4: You are confident in your assessment, but not absolutely certain. It is unlikely, but not impossible, that you did not understand some parts of the submission or that you are unfamiliar with some pieces of related work.

**Review Rating:**

6: Borderline Leaning Accept: Technically sound submission where reasons to accept outweigh reasons to reject. Please use sparingly.

**Review Summary:**

Good work on developing AutoML methods that will be available for R users.
More comparisons should be performed on more benchmarks in R.

Even if it is not fair, can we see what the results will be if we do this with some tool available in Python?

**Technical Quality And Correctness:**

The presented package is following a pipeline that consists of four steps.

1) Can you please explain how you select the methods implemented for data preprocessing? If you are using all of them or the user needs to specify what they want to use?

2) Why did you select DALEX for model interpretation? Is this a method that leads to state-of-the-art tree-based model explanations?

3) More comprehensive comparison with the H2O package should be performed on more datasets. Some statistical analysis should also be done to see what are the results. Part C in the appendix presents some training and testing accuracies that are one, is this implicitly the point that overfitting is happening?

4) What is the performance gain of using a pipeline from foster against H20 on each dataset? Can you estimate it?

5) Further analysis for regression should be performed.

---

### Official Review · Reviewer_xy9r · 2023-05-11

**Potential Impact On The Field Of Automl Rating:** 2
**Technical Quality And Correctness Rating:** 2
**Clarity:** Overall, I find the paper clear and e…
**Clarity Rating:** 4

**Summary Of Contributions:**

This paper introduce forester, a new library to perform tree-based AutoML on tabular data in R. It has a simple API that invokes the 4 steps in the pipeline in one call: (1) basic sanity checks on the data, (2) data preprocessing, (3) training and tuning of the model, and (4) performance evaluation. forester also comes with a report generator to provide users with more insights. At the end of the paper, the authors perform a comparison of forester with the H2O (another AutoML package in R) on a few binary classification dataset and show that forester is capable of building strong models.

**Actions Required To Increase Overall Recommendation:**

It would be nice if the author can:
- Explain what's novel in their forester work (H2O also supports wide range of metrics, so "we can compare models using the most commonly used metrics" isn't exactly novel
- See my other comments on the technical section

**Overall Review:**

Pros:
- The API of forester is very intuitive and simple to use
- It comes with a nice report generator
- It runs relatively well on the 8 binary classification datasets, comparing to H2O

Cons:
- forester is too limited in terms on the type of tasks, data type, and only support tree-based models
- More experiments are needed to to validate it's performance (ideally, more AutoML libraries should be included in the comparison as well)

**Potential Impact On The Field Of Automl:**

forester comes with a simple API and a pretty nice report generator, both of which are very friendly to machine learning newbie. However, I don't see a lot of novelty in this work comparing to other AutoML libraries in R. forester only supports binary classification and regression problems on tabular data, and is limited to tree-based approaches -- which have already been well-supported by libraries like H2O.

**Review Confidence:**

4: You are confident in your assessment, but not absolutely certain. It is unlikely, but not impossible, that you did not understand some parts of the submission or that you are unfamiliar with some pieces of related work.

**Review Rating:**

4: Weak Reject: For instance, a submission with minor technical flaws, and limited impact.

**Review Summary:**

The forester library is quite neat on the narrow type of task that it supports (binary classification and regression on tabular data), but it is a bit lacking in novelty comparing to existing AutoML library in R.

**Technical Quality And Correctness:**

The author has made forester available on GitHub and has put up a nice API documentation website for it. In the paper, it also includes a comparison of forester and the popular H2O package on 8 binary classification datasets. However, I have several questions regarding the experiment section:
1. The authors limited the maximum time of H2O package to be the same as the forester for each of the iteration. However, forester only supports tree-based methods which are generally very fast, whereas H2O supports more algorithms such as deep learning (fully connected neural net). As such, H2O might naturally take more time to finish since it has to spend time on these additional algorithms, and limiting the maximum run time could result in under-exploration. Can we either ensure that both libraries converged (by limiting to the typical 1 hr time), or at least exclude the non-tree-based algorithms when limiting H2O's total run time?
2. How's the 8 dataset selected? Is it manually picked or selected by random?
3. Would it be possible to provide some results on regression problems?